# SEAgent: Self-Evolving Computer Use Agent with Autonomous Learning from Experience

## Abstract

Repurposing large vision-language models (LVLMs) as computer use agents (CUAs) has led to substantial breakthroughs, primarily driven by human-labeled data. However, these models often struggle with novel and specialized software, particularly in scenarios lacking human annotations. To address this challenge, we propose SEAgent, an agentic self-evolving framework enabling CUAs to autonomously evolve through interactions with unfamiliar software. Specifically, SEAgent empowers computer-use agents to autonomously master novel software environments via experiential learning, where agents explore new software, learn through iterative trial-and-error, and progressively tackle auto-generated tasks organized from simple to complex. To achieve this goal, we design a World State Model for step-wise trajectory assessment, along with a Curriculum Generator that generates increasingly diverse and challenging tasks. The agent's policy is updated through experiential learning, comprised of adversarial imitation of failure actions and Group Relative Policy Optimization (GRPO) on successful ones. Furthermore, we introduce a specialist-to-generalist training strategy that integrates individual experiential insights from specialist agents, facilitating the development of a stronger generalist CUA capable of continuous autonomous evolution. This unified agent ultimately achieves performance surpassing ensembles of individual specialist agents on their specialized software. We validate the effectiveness of SEAgent across five professional software of OSWorld, ScienceBoard and AndroidWorld. Our approach achieves a significant improvement over a competitive open-source CUA, UI-TARS. All the code and models will be made publicly available to foster further research.

## 1 Introduction

*"A new generation of agents will acquire superhuman capabilities by learning predominantly from experience." (Silver & Sutton, 2025)*

*— David Silver, Richard S. Sutton*

With the rapid development of large vision-language models (LVLMs) (Touvron et al., 2023; Grattafiori et al., 2024; Bai et al., 2025; Wang et al., 2024; OpenAI, 2023; Anthropic, 2025b; Team et al., 2023), computer use agents (CUAs) (Anthropic, 2024; OpenAI, 2025; Qin et al., 2025; Lin et al., 2024; Wu et al., 2024b) have not only emerged but also demonstrated increasing practical utility. By leveraging the powerful perception and reasoning capabilities of LVLMs, these agents can interpret screenshots as visual inputs and operate computers via keyboard and mouse actions. Despite their promising capabilities, current CUAs (Qi et al., 2024; Putta et al., 2024; Deng et al., 2023; He et al., 2024; Bai et al., 2024; Lu et al., 2024) primarily depend on costly human-curated datasets (Deng et al., 2023; Chen et al., 2024; Wu et al., 2024b; Kapoor et al., 2024; Li et al., 2024), which are typically derived from demonstrations (Lu et al., 2024; Zhang & Zhang, 2023; Gur et al., 2023; Rawles et al., 2023; Zhang et al., 2024a) or video tutorials in the wild (Xu et al., 2024). However, new software continuously emerges and existing software may regularly be updated, often in the absence of annotated human data. It is both necessary and timely to enter an era that emphasizes learning from experience (Silver & Sutton, 2025) in CUA domain. In this paper, we aim to enable CUAs to autonomously explore unfamiliar software environments and evolve into experts without relying on human supervision.

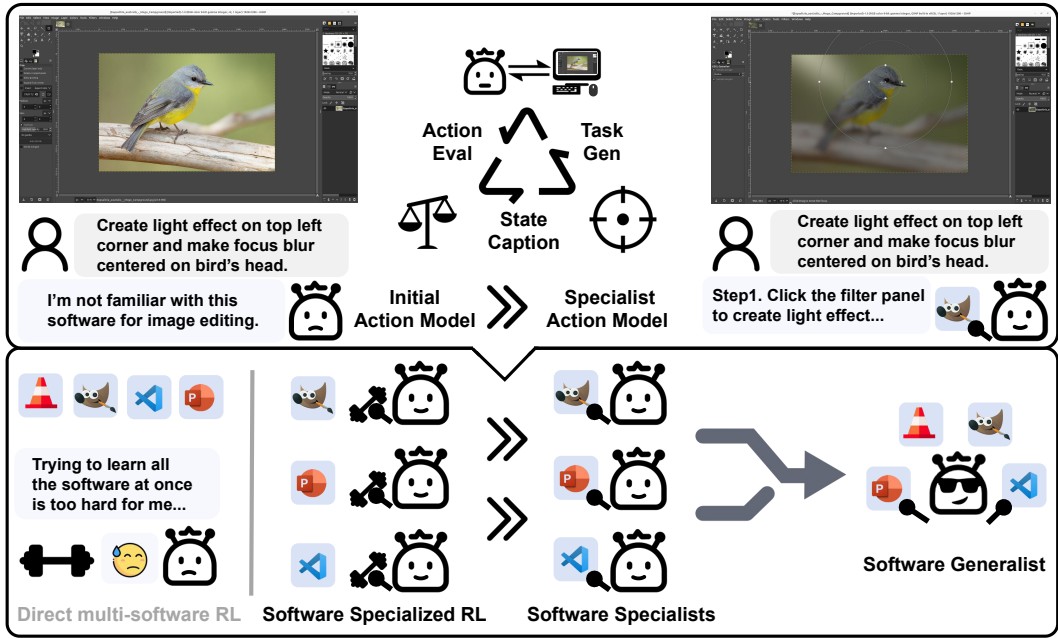

Figure 1: **SEAgent enables computer use agents self-evolving in novel environments** by autonomously exploring and learning from their own experiences without human intervention. The specialist-to-generalist training strategy further enhances the development of a strong generalist agent.

To address this challenge, we propose SEAgent, an agentic self-evolving framework in which Computer Use Agents (CUAs) are exposed to previously unfamiliar software environments and engage in autonomous exploration and experiential learning, as illustrated in Fig. 1. Enabling such self-evolution requires addressing two key challenges: (1) generating executable tasks within unfamiliar software environments, and (2) accurately assessing task success and precisely identifying the step at which failure occurs. To this end, we introduce a **World State Model** for environmental state captioning and step-wise trajectory assessment, together with a **Curriculum Generator** powered by a continuously updated software guidebook memory to generate increasingly diverse and challenging tasks, thereby establishing a curriculum learning paradigm. The agent's policy is optimized through experiential learning from both failures and successes, combining adversarial imitation of failure actions and Group Relative Policy Optimization (GRPO) on successful ones.

Given the critical role of reward accuracy, we conduct extensive evaluations and observe that existing reward models of computer use tasks fall short in terms of judgment precision and reward density. Leveraging the enhanced long-context processing capabilities of advanced LVLMs, we input the agent's full trajectory of states into the reward model and fine-tune a reward model, World State Model, using Qwen2.5-VL (Bai et al., 2025), substantially narrowing the gap with commercial models such as GPT-4o (OpenAI, 2023) with +7.5% improvement in precision compared to baseline model in evaluating CUAs' trajectories on AgentRewardBench (Lù et al., 2025), enable World State Model to provide high quality step level reward signals in self-evolving agentic system.

Moreover, SEAgent enables agents to evolve into either single-software specialists or multi-software generalists. To overcome the limitation that directly training a generalist underperforms compared to specialists, inspired by (Zhang et al., 2024c), we introduce a novel specialist-to-generalist training strategy, which even surpasses the performance of individual specialists on their respective software applications. We perform extensive experiments of SEAgent built on UI-TARS (Qin et al., 2025) and evaluated on five professional software applications from OSWorld (Xie et al., 2024). SEAgent with the specialist-to-generalist strategy significantly improves the UI-TARS (Qin et al., 2025). Furthermore, SEAgent with the specialist-to-generalist strategy outperforms both specialist RL and generalist RL by a substantial margin, demonstrating the effectiveness of the specialist-to-generalist paradigm. We also validate this strategy on UI-TARS-1.5 on ScienceBoard (Sun et al., 2025) on out of domain scientific softwares. In general, SEAgent offers a promising approach for developing more powerful and versatile computer-use agents without human involvement.

## 2 RELATED WORK

**Agent for Computer Use.** With recent revolution in LLM and LVLMs (Touvron et al., 2023; Grattafiori et al., 2024; Liu et al., 2023a; Bai et al., 2025; Wang et al., 2024), processing human level perception and reasoning ability, building computer use agent is also intensively studied (Hu et al., 2024; Hong et al., 2024; Cheng et al., 2024; Nguyen et al., 2024; Lin et al., 2024). These agents either takes only text input from structured text (Qi et al., 2024) or more like human, take screenshot and text condition as multi-modal input. Although intensively studied and perform well on in-domain benchmark (Lu et al., 2024; Zheng et al., 2024; Liu et al., 2024; Li et al., 2025; Cheng et al., 2024), The computer use agent still fall largely behind human level performance in simulation environment (Xie et al., 2024; Rawles et al., 2024; Koh et al., 2024; Zhou et al., 2023), as its challenge the multi-dimension ability of LVLMs in grounding, decision making and reasoning with works done breaking this process into different expert models (Gou et al., 2024; Wan et al., 2024) with agent calloberation (Agashe et al., 2024; 2025; Liu et al., 2023b; Zhang et al., 2025) through prompt engineering (Yan et al., 2023; He et al., 2024; Zhang et al., 2024b; Wang et al., 2023; Wu et al., 2024a), However, these training free methods improvements is restricted without fine-tuning. In this work, we dive into the next step of CUA where the pretrained agent is fine-tuned to learn from its own experience and achieves self-evolution on specialized novel software without human annotations.

**Reinforcement Learning for LLM/ LVLMs.** Previous post-training for LLM/ LVLMs (Touvron et al., 2023; Grattafiori et al., 2024; Liu et al., 2023a; Bai et al., 2025; Wang et al., 2024) mainly from supervised fine-tuning (SFT) (Liu et al., 2023a; Wei et al., 2022). Similar to imitation learning in RL, SFT teach model to output desired output. This makes SFT highly dependent on high quality human procedure data. Recently, DeepSeek-R1 (Guo et al., 2025) achieve strong reasoning ability through Group Relative Policy Optimization (GRPO) (Shao et al., 2024) with verifiable rewards. Previous works (Ouyang et al., 2022; Ziegler et al., 2019; Rafailov et al., 2023) also apply RL to single turn optimization from human feedback. However, in agentic applications like computer use where environment feedback is sparse, where success is achieved with multi-step interactions. It is important to introduce stable step level reward signals. Previous works on RL for agent (Bai et al., 2024; Qi et al., 2024; Zhou et al., 2024; Zhai et al., 2024; Carta et al., 2023) fine-tune their own critic model for advantage estimation based on output reward model (ORM) or use DPO (Rafailov et al., 2023) policy updates based on interaction data (Putta et al., 2024; Qin et al., 2025). In this work, we dive into evaluation of different strategies for building the best performing reward model for CUAs and find that full process based analysis provide the most accurate results compared to training specific critic model to perform advantage estimation in (Bai et al., 2024; Qi et al., 2024).

## 3 METHODS

**Problem Formulation.** The objective of SEAgent is to establish a training pipeline enabling the Computer Use Agent (CUA) to autonomously explore its environment (Sec. 3.1) and progressively self-evolve on novel software applications via reinforcement learning from experience (Sec. 3.2). Specifically, the SEAgent pipeline comprises three primary components: an Actor Model $\pi$ performing exploratory actions to accomplish these tasks, and a World State Model $\mathcal{M}_{state}$ describing the current environment state and evaluating the success or failure of executed actions, and a Curriculum Generator $\mathcal{M}_{task}$ that continuously proposes more diverse and challenging exploration tasks:

**(1) Actor Model $\pi$:** The policy $\pi(a|s_t, I)$ defines the probability of taking action $a$ at time step $t$, conditioned on the current state $s_t$ and the overall task instruction $I$.

**(2) World State Model $\mathcal{M}_{state}$:** This component is a fine-tuned Large Vision-Language Model (LVLM) responsible for providing detailed descriptions of environment states. It also evaluates each step of the trajectory executed by the Actor Model $\pi$, producing trajectory judgement $\mathcal{J}$ which indicates whether the task has been successfully completed. Joint training with state change captioning $\mathcal{C}$ of the software GUI has been shown to enhance judgment accuracy, as shown in Table 1.

**(3) Curriculum Generator $\mathcal{M}_{task}$:** This component utilizes a powerful Large Language Model (LLM) to automatically generate novel exploration tasks. It also maintains and updates a software guidebook $U$ based on the state change captioning $\mathcal{C}$ and the trajectory judgement $\mathcal{J}$ provided by

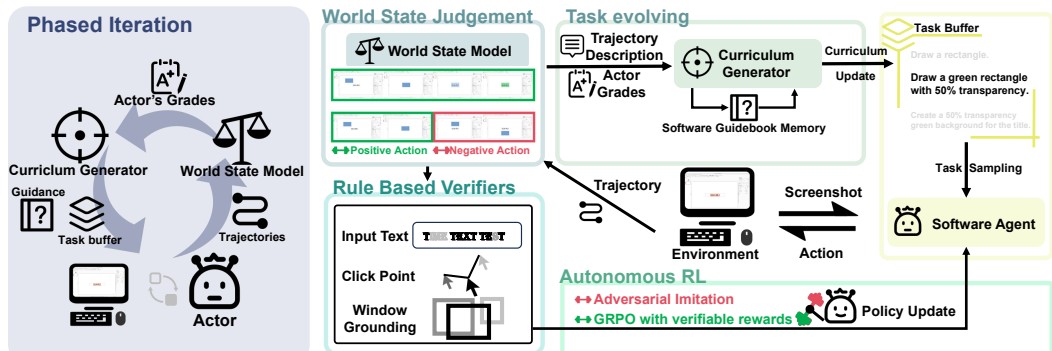

Figure 2: **SEAgent autonomous exploration and experiential learning pipeline.** Guided by tasks generated by the Curriculum Generator, the Actor Model is updated according to step-level rewards from the World State Model through verifiable reward functions tailored for different action types.

$\mathcal{M}_{state}$ during interactions. The gradually enriched guidebook $U$ enables $\mathcal{M}_{task}$ to progressively generate increasingly diverse and challenging tasks in a curriculum learning fashion.

SEAgent can be applied to enable the self-evolution of a computer-use agent, either as a specialist for a single software or as a generalist across multiple software. However, we observe that direct training for generalist agents is suboptimal. We introduce a specialist-to-generalist training strategy, which achieves improved overall performance than training multiple generalist agents detailed in Sec. 3.3.

### 3.1 Autonomous Exploration with Self-evolving Curriculum

Autonomous exploration is essential for enabling the Computer Use Agent (CUA) to develop proficiency in novel software applications that are previously unseen or poorly understood. This process involves addressing two key challenges: (1) generating executable tasks within unfamiliar software environments, and (2) evaluating task completion success and pinpointing the specific step at which failure occurs. To tackle these challenges, we introduce two novel components: the World State Model $\mathcal{M}_{state}$ and the Curriculum Generator $\mathcal{M}_{task}$. These components jointly support a **self-evolving curriculum paradigm**, which facilitates the autonomous generation of increasingly diverse and challenging tasks.

The **self-evolving curriculum paradigm** pipeline is structured into $P$ sequential phases. Before the first phase, a set of initial tasks targeting basic GUI operations is generated (details provided in Sup. C.1). In each phase, these tasks are executed and step-wise evaluated. The resulting judgments and descriptions of the exploration trajectories are fed into the Curriculum Generator $\mathcal{M}_{task}$, which updates a self-maintained software guidebook $U$. Leveraging this updated guidebook and the current capabilities of the CUA, the generator then produces more diverse and challenging tasks for subsequent phases. The following outlines each step of the process in detail:

**(1) Task initiation:** The initial state of the unfamiliar software is provided, typically in the form of screenshots of its basic GUI interface. The World State Model $\mathcal{M}_{state}$ performs dense captioning of the GUI elements, including button detection and OCR-based recognition. These detailed captions are passed to the Curriculum Generator $\mathcal{M}_{task}$, which generates an initial set of task instructions $\mathcal{I}_0 = \{I_0^{(1)}, I_0^{(2)}, \cdots\}$ along with an initial software guidebook $U_0$ for the software.

**(2) World state judgment:** In the $p$-th phase of *Auto Exploration*, the Actor Model $\pi_p$ executes tasks based on the instructions in $\mathcal{I}_p$. Each execution is evaluated by the World State Model $\mathcal{M}_{state}$, which provides feedback $\mathcal{J}_p = \{J_p^{(1)}, J_p^{(2)}, \cdots\}$ for each step within the operation trajectory. In addition, it generates a detailed description of GUI state changes based on captured screenshots, denoted as $\mathcal{C}_p$.

**(3) Task self-evolving:** Based on the outcomes $\mathcal{J}_p$ and $\mathcal{C}_p$, the Curriculum Generator $\mathcal{M}_{task}$ produces a more challenging task set $\mathcal{I}_{p+1}$ and expands the agent's knowledge boundary by updating the software guidebook to $U_{p+1}$. The detailed prompting process is illustrated in Fig. 9. This iterative update can be formalized as:

$$U_{p+1}, \mathcal{I}_{p+1} = \mathcal{M}_{task}(U_p, \mathcal{I}_p, \mathcal{J}_p, \mathcal{C}_p) \tag{1}$$

Here, $U_{p+1}$ serves as a more comprehensive software guidebook memory, while $\mathcal{I}_{p+1}$ represents a more challenging task set tailored to the current capabilities of the Actor Model $\pi_p$. Examples of $\mathcal{I}_p$ are provided in Fig. 4, where the Actor Model $\pi$ demonstrates curriculum learning by handling increasingly complex tasks across different phases $p$. Illustrations of $U_p$ across various software applications are provided in Sup. J. Comparison with previous methods (Murty et al., 2025; 2024; Sun et al., 2024) on task generation are detailed in Sup.C.2

**(4) Autonomous RL Training:** Through iterative reinforcement learning, the Actor Model $\pi_p$ is updated based on its execution of the instruction set $\mathcal{I}_p$, guided by evaluation feedback $\mathcal{J}_p$ and a set of action-specific verifiable functions $\mathcal{R}_{\text{verifer}}$. The resulting policy $\pi_{p+1}$ is then used as the actor in the subsequent phase. Further details are provided in Sec. 3.2.

## 3.2 Reinforcement Learning from Experience

The World State Model $\mathcal{M}_{state}$ provides step-level reward signals for reinforcement learning. Unlike previous reward models for CUA (Qi et al., 2024; Bai et al., 2024; Putta et al., 2024; Pan et al., 2024; Lù et al., 2025), our $\mathcal{M}_{state}$ model takes the entire trajectory of states and actions, $\mathcal{H} = \{(s_0, a_0), (s_1, a_1), \ldots\}$, as input. It classifies each action $a$ as either $a_F$ or $a_T$, where $a_F$ indicates an incorrect action leading to failure or redundant loops, and $a_T$ represents a correct action that contributes to successful progression without redundancy. The curated prompt used for judgment is depicted in Fig. 8 with input/output format detailed in Sec.**??**. For historical states that result in $a_T$, we encourage CUA to reinforce these actions through verifiable rewards defined by a set of functions $\mathcal{R}_{\text{verifer}} = \{r_{dist}\}$. Conversely, for states leading to $a_F$, we penalize them using negative KL divergence with adversarial imitation.

**Adversarial Imitation for Failure Action Punishment.** To explicitly encourage the policy to diverge from failure-inducing behaviors, we employ a contrastive log-ratio loss based on a reference failure action $a_F$. This objective encourages the policy to sample actions $a$ that minimize alignment with the failure action $a_F$:

$$\mathcal{L}_{\text{AI}}(\pi_\theta) = \mathbb{E}_\nu \left[ -\log \frac{\pi_\theta(a \mid s, I)}{\pi_{\text{ref}}(a_F \mid s, I)} \right] \tag{2}$$

**Verifiable Rewards for Correct Action Encouragement.** To more effectively guide the policy toward correct actions $a_T$, we adopt Reinforcement Learning with Verifiable Rewards (RLVR) (Guo et al., 2025; Shao et al., 2024), which has recently shown success in enhancing language models on tasks with objectively verifiable answers, such as mathematics (Shao et al., 2024), and more recently, counting and grounding in the vision-language domain (Liu et al., 2025; Shen et al., 2025; Meng et al., 2025). After labeling the correct step $(s, a_T)$ using the World State Model, we apply Group Relative Policy Optimization (GRPO), computing the relative advantage of each response based on its reward:

$$A^{(i)} = \frac{r^{(i)} - \text{mean}(\{r^{(j)}\}_{j=1}^G)}{\text{std}(\{r^{(j)}\}_{j=1}^G)}, \quad i = 1, \cdots, G. \tag{3}$$

As we design distinct reward signals for different action types, we define the reward function between a predicted action $a$ and the ground-truth action $a_T$ as:

$$r^{(i)} = r(a^{(i)}, a_T) = \mathbb{I}\left(\text{type}(a^{(i)}) = \text{type}(a_T)\right) + r_{\text{dist}}(a^{(i)}, a_T), \tag{4}$$

where $\mathbb{I}(\cdot)$ is the indicator function that returns 1 if the predicted action and ground-truth action are of the same type, and 0 otherwise. The distance-based reward term $r_{\text{dist}}(a^{(i)}, a_T)$ is defined according to the specific action type: for `click` actions, it is computed based on the normalized L1 distance between the clicked coordinates; for `drag` and `select` actions, it is computed using the Intersection over Union (IoU) between the predicted and ground-truth bounding boxes; and for `type` actions, it is determined by the character-level BLEU score between the predicted and ground-truth text. All $r_{\text{dist}}$ rewards are normalized to the range $[0, 1]$ to ensure consistency across different action types. A comprehensive list of $r_{\text{dist}}(a^{(i)}, a_T)$ definitions for various action types is provided in Tab. 9. The final loss of GRPO is directly adopted from (Shao et al., 2024):

$$\mathcal{L}_{\text{GRPO}}(\pi_\theta) = -\mathbb{E}_{(s,I)\sim\mathcal{D},\{a^{(i)}\}_{i=1}^{G}\sim\pi_{\text{ref}}(\cdot|s,I)} \tag{5}$$

$$\left[ \frac{1}{G} \sum_{i=1}^{G} \frac{1}{|a^{(i)}|} \sum_{t=1}^{|a^{(i)}|} \left\{ \min\left( r_t^{(i)}(\theta) A^{(i)}, \text{clip}(r_t^{(i)}(\theta), 1-\epsilon, 1+\epsilon) A^{(i)} \right) - \beta\, D_{\text{KL}}^{(i,t)}(\pi_\theta \| \pi_{\text{ref}}) \right\} \right],$$

where $r^{i,t}(\theta) = \dfrac{\pi_\theta(a^{(i)}|s,I)}{\pi_{\theta_{\text{ref}}}(a^{(i)}|s,I)}$ and $D_{\text{KL}}^{i,t}(\pi_\theta, \pi_{\text{ref}}) = \dfrac{\pi_{\text{ref}}(a^{(i)}|s,I)}{\pi_\theta(a^{(i)}|s,I)} - 1 - \log\dfrac{\pi_{\text{ref}}(a^{(i)}|s,I)}{\pi_\theta(a^{(i)}|s,I)}.$

Similar to (Shao et al., 2024; Guo et al., 2025), advantage $A$ is weighted on the whole reasoning token logits to encourage free form thinking for performing action and planning.

The final training loss is defined as a weighted combination of positive and negative action samples, i.e., correct actions $a_T$ and incorrect actions $a_F$: $\mathcal{L}(\pi(\theta)) = \mathcal{L}_{\text{GRPO}} + \gamma \mathcal{L}_{\text{AI}}$. We set $\gamma = 0.2$ during training, and the rationale for this choice is discussed in the ablation study presented in Sup. F.

This strategy is shown to be more effective in Sec. 4.2 compared to Generalized Advantage Estimation (GAE) (Schulman et al., 2015)-based RL methods (Qi et al., 2024; Bai et al., 2024), as the more powerful reward model $\mathcal{M}_{state}$ provides accurate step-level reward signals by leveraging the entire episode trajectory $\mathcal{H}$ from a global perspective.

### 3.3 FROM SPECIALIST TO GENERALIST.

Achieving a generalist agent capable of operating across multiple software platforms is an ambitious and valuable goal. We first attempted to train such a generalist directly using the proposed SEAgent framework across all software environments. However, this approach led to suboptimal performance compared to specialized agents, as the actor struggled to learn effectively in the multi-software environment.

We thus introduce a specialist to generalist strategy, as illustrated in Fig. 1. Specifically, we first train software-specialized agents via SEAgent on individual environments, allowing each to master a specific application. These specialists are then distilled into a single generalist model through supervised fine-tuning (SFT) on synthesized successful trajectories. Finally, the generalist is refined via SEAgent on multiple software. This generalist, now equipped with better reasoning, planning abilities, and software-specific commonsense, achieves significantly improved performance, outperforming both the SEAgent via direct general RL and the performance combination of multi-specialists as in Table 2.

## 4 EXPERIMENTS

### 4.1 EVALUATION OF REWARD MODEL FOR COMPUTER USE AGENT.

Providing CUA agents with reliable reward signals is crucial for enabling self-evolution. Building on AgentRewardBench (Lù et al., 2025), which focuses on web tasks, we extend the evaluation to a broader set of PC software environments. Specifically, we evaluate on all 339 feasible tasks from OSWorld (Xie et al., 2024). Trajectories are sampled from UI-TARS (Qin et al., 2025) and Gemini-2.5-Pro (Google DeepMind, 2025), and a rule-based evaluation is used as ground-truth supervision to compute a confusion matrix for each reward model's predictions.

The judging strategy in AgentRewardBench (Lù et al., 2025) is limited, as it relies solely on the final state and action history. It is more natural and reliable for a judge model to consider the entire trajectory when assessing task success. For example, a final state message like "Your flight ticket has been successfully booked" does not confirm whether the correct date and time were selected, which can lead to compromised judgment accuracy.

However, we observe that current open-sourced LVLMs perform poorly under this more holistic evaluation strategy. As shown in Fig. 3, feeding additional historical screenshots into Qwen2.5-VL (Bai et al., 2025) significantly degrades its Average Precision (AP), diverging notably from GPT-4o (Hurst et al., 2024) on the same curated prompt. We attribute this performance drop to the

Table 1: **Precision and Negative Predictive Value (NPV)** on AgentReardBench (Lù et al., 2025) and OSWorld (Xie et al., 2024) with last screenshot only (LS) or entire process screenshots (ES) as input. World State Model closes the gap with commercial model. The co-training with screenshot change description (CD) improves judgment precision.

| Model | Input | AgentRewardBench | | OS-World-Full | | Prof/Office | |
|---|---|---|---|---|---|---|---|
| | | Precision | NPV | Precision | NPV | Precision | NPV |
| GPT-4o (Hurst et al., 2024) | LS | 68.1 | 92.3 | 46.3 | 88.2 | 40.5 | 81.0 |
| | ES | 72.1 | 92.2 | 74.6 | 95.2 | 70.4 | 85.3 |
| Qwen2.5-VL-72B (Bai et al., 2025) | LS | 64.5 | 94.2 | 41.5 | 86.9 | 31.7 | 78.7 |
| | ES | 26.2 | 83.0 | 26.8 | 83.0 | 25.6 | 76.6 |
| Qwen2.5-VL-7B (Bai et al., 2025) | LS | 64.1 | 90.3 | 37.3 | 85.2 | 31.8 | 79.0 |
| | ES | 25.4 | 83.8 | 20.0 | 81.7 | 23.5 | 76.0 |
| World State Model (w/o CD) | ES | 69.1 | 88.5 | 71.1 | 88.4 | 65.0 | 81.1 |
| World State Model (w/ CD) | ES | 71.6 | 91.2 | 73.9 | 90.5 | 69.3 | 82.0 |

insufficient pretraining of Qwen2.5-VL on long sequences of high-resolution screenshots, which pushes it toward the limits of its 32K context length.

To address this, we propose World State Model, a distilled model based on Qwen2.5-VL-7B. The training process for World State Model uses a dataset of 0.86K GPT-4o (Hurst et al., 2024) generated evaluations on trajectories with dense GUI change descriptions, exclusively from the Chrome browser within the OSWorld (Xie et al., 2024) environment. Alongside the primary judgment task, we also find it effective to co-training the model with change description (CD) task for describing the difference of the screenshot before and after an action. Training data and settings are detailed in Sup. A. Despite being trained solely on Chrome data, World State Model exhibits strong generalization to other professional software in OSWorld and to the external AgentRewardBench (Lù et al., 2025) benchmark. This demonstrates that the model learns transferable judgment patterns rather than overfitting to a single application.

As evaluated in Tab. 1 and further analyzed in Fig. 3, World State Model achieves state-of-the-art performance among open-sourced models, significantly narrowing the gap with GPT-4o (Hurst et al., 2024). Despite being trained on a relatively small dataset, World State Model is explicitly encouraged to capture the sequential dependencies among historical screenshots and to perform step-by-step reasoning for final judgment. It provides reliable, step-level reward signals that support downstream policy learning (Training reward w.r.t. different reward signal providers is depicted in Fig. 5), allowing our agentic system to evolve using fully open-sourced models while avoiding inefficient and costly API calls to proprietary models.

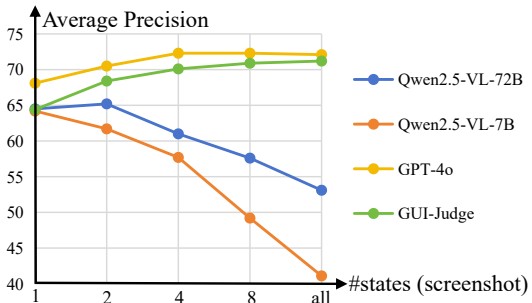

Figure 3: **The Average Precision on AgentRewardBench (Lù et al., 2025)**, where GUI-Judge exhibits an improvement in AP as the number of input middle states increases, showing a similar trend to that of the closed sourced GPT-4o.

## 4.2 SELF EVOLUTION OF GUI AGENTS

**Models Before Self-Evolution.** Our self-evolving system is initialized with three locally deployed models: UI-TARS-7B-DPO (Qin et al., 2025) as the Actor Model, World State Model as the step-level reward model, and Qwen2.5-72B (Yang et al., 2024) as the Curriculum Generator. As shown in Tab. 2, the initial Actor Agent achieves an average success rate of 21.5% across five professional software applications from OSWorld.

**Evolution Process Details.** The evolution process begins with the Curriculum Generator producing an initial instruction set ($\mathcal{I}_0$), averaging 150.2 instructions. The Actor Model executes these tasks, and the resulting trajectories are evaluated by World State Model and parsed into an average of 1361.5 multi-turn conversation pairs (detailed statistics are in Sup.H). We then perform reinforcement fine-tuning (RFT) for 1k iterations on 8 NVIDIA A100 80GB GPUs, with a batch size of 16 and a learning rate of $2 \times 10^{-5}$ scheduled via cosine decay. This process is repeated for three phases.

Table 2: **Success Rate (SR) on OSWorld (Xie et al., 2024)**. SEAgent demonstrates strong performance after reinforcement learning. In addition to evolving on separate software, a new General Model achieves better performance after another iteration of SEAgent. *Indicates specialist agents trained separately for each software with ensembled results. All results are averaged over five runs.

| Model | VScode | GIMP | Impress | VLC | Writer | Overall |
|---|---|---|---|---|---|---|
| Human Performance | 73.9 | 73.1 | 80.9 | 70.6 | 73.9 | 74.5 |
| GPT-4o (Hurst et al., 2024) | 4.35 | 3.85 | 6.77 | 16.1 | 4.35 | 7.08 |
| GPT-4V (OpenAI, 2023) | 0.00 | 7.69 | 2.52 | 18.3 | 4.35 | 6.59 |
| Gemini-Pro-1.5 (Team et al., 2023) | 0.00 | 11.5 | 13.2 | 6.53 | 8.71 | 7.99 |
| Claude3.7 Sonnet (Anthropic, 2025a) | 18.8 | 24.4 | 10.6 | 27.5 | 17.4 | 19.7 |
| Gemini-Pro-2.5 (Google DeepMind, 2025) | 21.7 | 26.9 | 9.92 | 25.5 | 24.6 | 21.7 |
| UI-TARS-7B-DPO (Lu et al., 2024) | 30.4 | 34.6 | 17.0 | 11.8 | 13.6 | 21.5 |
| UI-TARS-72B-DPO (Lu et al., 2024) | 39.1 | 53.8 | 23.4 | 15.3 | 26.1 | 31.5 |
| DigiRL (Bai et al., 2024) (Specialized RL)* | 43.7 | 45.4 | 19.6 | 25.0 | 19.1 | 30.6 |
| WebRL (Qi et al., 2024) (Specialized RL)* | 36.5 | 37.7 | 20.4 | 29.4 | 21.7 | 29.1 |
| SEAgent (Specialized RL)* | 46.1 | 50.0 | 21.3 | 31.8 | 33.0 | 36.4 |
| DigiRL (Bai et al., 2024) (General RL) | 38.3 | 46.2 | 19.1 | 25.9 | 19.1 | 29.7 |
| WebRL (Qi et al., 2024) (General RL) | 35.6 | 33.1 | 18.7 | 27.0 | 15.7 | 26.0 |
| SEAgent (General RL) | 40.8 | 42.3 | 21.7 | 28.2 | 30.4 | 32.6 |
| SEAgent (General SFT) | 36.5 | 41.5 | 25.5 | 30.6 | 32.2 | 33.3 |
| SEAgent (Specialist-to-Generalist) | **47.8** | **50.8** | **29.8** | **35.3** | **36.5** | **40.0** |

Table 3: **Success Rate (SR)** on OSWorld (Xie et al., 2024) and ScienceBoard (Sun et al., 2025).

| Benchmark | OSWorld (Xie et al., 2024) | | | | | ScienceBoard (Sun et al., 2025) | | | |
|---|---|---|---|---|---|---|---|---|---|
| *Software* | Impress | Writer | GIMP | VScode | VLC | ChamerX | GrassGIS | KAlgebra | Celestia |
| UI-TARS-1.5-7B-DPO | 29.8 | 39.1 | 51.5 | 60.9 | 23.5 | 12.4 | 0.0 | 11.6 | 4.9 |
| UI-TARS-1.5-7B-DPO + SEAgent | 31.9 | 43.5 | 56.9 | 60.9 | 35.3 | 31.0 | 20.6 | 29.0 | 15.2 |

**Specialist Evaluation.** For a fair comparison with previous methods (Bai et al., 2024; Qi et al., 2024), we train specialist agents for five different software applications. We adapt their strategies by initializing a separate critic model from UI-TARS-7B with randomly initialized MLP layers to regress value predictions using Generalized Advantage Estimation (GAE) (Schulman et al., 2015). As shown in Tab. 2 and Fig. 4, SEAgent, achieves superior performance. We attribute this to World State Model providing fine-grained, step-level rewards from the full history, which is more effective than relying on a separate critic to estimate advantages from sparse, final success/failure signals. Experimental results on mobile use GUI are supplied in Sec. D. We also provide comparison with previous task generation methods (Murty et al., 2025; Qi et al., 2024) on task generation are detailed in Sup.C.2.

As shown in Fig. 4 and Tab. 2, we train separate actor agents for five different software applications. Our approach, denoted as SEAgent (Specialist), achieves strong performance compared to previous reinforcement learning methods such as DigiRL (Bai et al., 2024) and WebRL (Qi et al., 2024). We attribute this improvement to the use of World State Model, which provides fine-grained, step-level reward signals derived from a comprehensive understanding of the full history of states and actions. This contrasts with previous approaches that rely on separate critic models—typically initialized from the actor itself—to estimate advantages from sparse, final success/failure signals. Furthermore, the curriculum of task instructions generated by the Curriculum Generator, as illustrated in Fig. 4, validates the effectiveness of our autonomous learning framework. These tasks progress from simple to complex based on the actor's evolving capabilities, enabling it to gradually specialize in each target software environment. Based on the observed evolution curves, we set the number of training phases to three, as performance gains saturate beyond that point.

**From Specialist to Generalist.** After training five strong software specialists, we pursue generalization. We collect task instructions from each specialist's training and use them to generate 3.5K successful trajectories. These trajectories, along with their reasoning traces, are distilled into a new base model (UI-TARS-7B) via supervised fine-tuning (SFT). This distilled model is then further optimized through RL across all five software environments. As shown in Tab. 2, the resulting generalist model surpasses the performance of the individual specialist ensemble.

**Results based on UI-TARS-1.5 and ScienceBoard.** Our work focuses on enabling agents to adapt to out-of-domain (OOD) and novel software where human-labeled data is not available. We applied our SEAgent pipeline to the UI-TARS-1.5 (Qin et al., 2025) using the same process described above

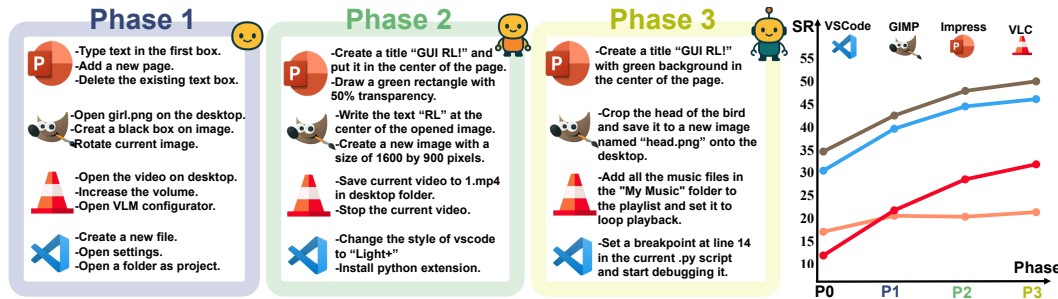

Figure 4: **Self-evolved task instructions and success rate (SR) curves across different software.** Tasks are progressively upgraded by the Curriculum Generator without human intervention, based on the evolving capabilities of the Actor Model at different training phases.

on two distinct benchmarks. As reported in Tab. 3, on OSWorld (Xie et al., 2024), we observed moderate performance gains. We hypothesize this is because UI-TARS-1.5's training data already cover OSWorld software environments, making it a familiar, in-domain evaluation for the base model. However, on the ScienceBoard (Sun et al., 2025) benchmark—a suite of scientific applications that are truly novel to UI-TARS-1.5—our pipeline delivers significant and substantial improvements. This strongly validates our core claim: SEAgent is most impactful when performing self-evolution learning on truly OOD software. We excluded two of the six ScienceBoard applications—Lean and TeX—as they are primarily text- and code-based software for mathematics and typesetting, which are not suitable for evaluating a GUI-centric agent like UI-TARS.

**Ablation Study of Specialist Training.** Our work focuses on enabling agents to adapt to out-of-domain (OOD) software. To test this, we applied our SEAgent pipeline to the UI-TARS-1.5 model. On OSWorld, an in-domain environment, we observed moderate gains. However, on the ScienceBoard (Sun et al., 2025) benchmark—a suite of scientific applications novel to the model—our pipeline delivered significant improvements. We excluded two Science-Board applications (Lean and TeX) as their text- and code-based interfaces are unsuitable for a GUI-centric agent like UI-TARS.

Table 4: Ablation of different configurations and their corresponding VScode success rates on OSWorld (Xie et al., 2024). Using World State Model as the reward model yields significant performance gains. We further compare different training strategies including supervised fine-tuning (behavior cloning), GRPO, and Adversarial Imitation (AI).

| Qwen2.5VL-72B | World State Model | SFT (BC) | GRPO | AI | VScode SR |
|---|---|---|---|---|---|
| | | | | | 30.4 |
| ✓ | | ✓ | | | 26.1 |
| ✓ | | | ✓ | | 28.3 |
| | ✓ | ✓ | | | 34.8 |
| | ✓ | ✓ | | ✓ | 39.1 |
| | ✓ | | ✓ | | 43.5 |
| | ✓ | | ✓ | ✓ | 46.1 |

## 5 CONCLUSION

In this work, we introduce SEAgent, an autonomous Computer Use Agent (CUA) exploration system that learns from its own experience on specific software. Powered by a robust World State Model that provides step-level reward signals, and a carefully designed reinforcement learning framework that encourages free-form reasoning through trial and error, the CUA is able to evolve into a specialist for individual software platforms. Furthermore, a specialist-to-generalist training strategy enables the development of a strong generalist agent capable of operating across multiple software environments. Given that computer software constitutes a highly regularized virtual world, we believe this work can inspire future research on agentic systems in both gaming and real world embodied environments.

**Limitations and future work.** While promising, our work still has several unresolved limitations. Firstly, our self evolving agent system is bounded by GUI-Judge to provide reliable reward signal instead of real signal from environment. As its still challenging to learning from sparse reward signal in complex environment. Secondly, though we tested on relatively complex and novel software like libreoffice-tools and GIMP. The task is still relatively simple as it only takes a human expert less than 20 step to accomplish. How to adapt the system to achieve hours-long workflow in even more challenging software used by real human expert are thus interesting future directions.

ETHICS STATEMENT

Our work adheres to the ICLR Code of Ethics. Our goal is to develop versatile computer-use agents that can autonomously adapt to new software, thereby automating a wide range of human workflows. We acknowledge that self-evolving agents learning without direct human oversight raises important safety considerations. To address this, our framework confines learning to isolated virtual machine environments and guides the agent's exploration through a structured curriculum and an automated reward model, preventing the acquisition of harmful or unintended behaviors. Potential for societal bias exists in the foundational models we use (e.g., UI-TARS (Qin et al., 2025), Qwen2.5-VL (Bai et al., 2025)) and could be inherited by our fine-tuned World State Model reward model, World State Model, and Curriculum Generator. A significant ethical benefit of our approach is its ability to bypass the need for costly, human-curated datasets, thus reducing the reliance on intensive manual annotation labor. We acknowledge the computational resources required for this form of experiential learning and are committed to the responsible development of capable and safely-evolving AI agents.

REPRODUCIBILITY STATEMENT

To ensure full reproducibility and to contribute to the community, we firmly commit to **open-sourcing our entire project** after the peer-review process. This includes **all source code** for our agentic self-evolving framework (SEAgent), encompassing the World State Model, Curriculum Generator, and our implementations of the learning algorithms. Furthermore, we will release **all model weights**, including our fine-tuned World State Model reward model and all specialist and generalist agents trained with our specialist-to-generalist strategy. Our work builds on public models like UI-TARS-1.5 (Qin et al., 2025) and Qwen2.5-VL (Bai et al., 2025), for which we provide exact identifiers. The appendix will offer a comprehensive guide to the experimental setup, detailing software configurations for the OSWorld (Xie et al., 2024) and ScienceBoard (Sun et al., 2025) and Android-World (Zhang et al., 2024b) benchmarks, all training hyperparameters, and the computational resources required to fully replicate our findings.

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

## DISCLOSURE ON THE USE OF LLMS

During the final drafting stages of this paper, we consulted Large Language Models (LLMs) to improve the manuscript's clarity and linguistic precision. The LLM served as an advanced editing tool, providing suggestions on syntax, word choice, and overall readability for the author-written text. We emphasize that this was an iterative process where the authors directed the tool and made all final decisions regarding the text. No part of the paper's core scientific arguments, methodology, or results was generated by the LLM. The authors bear full responsibility for all content and claims presented herein.

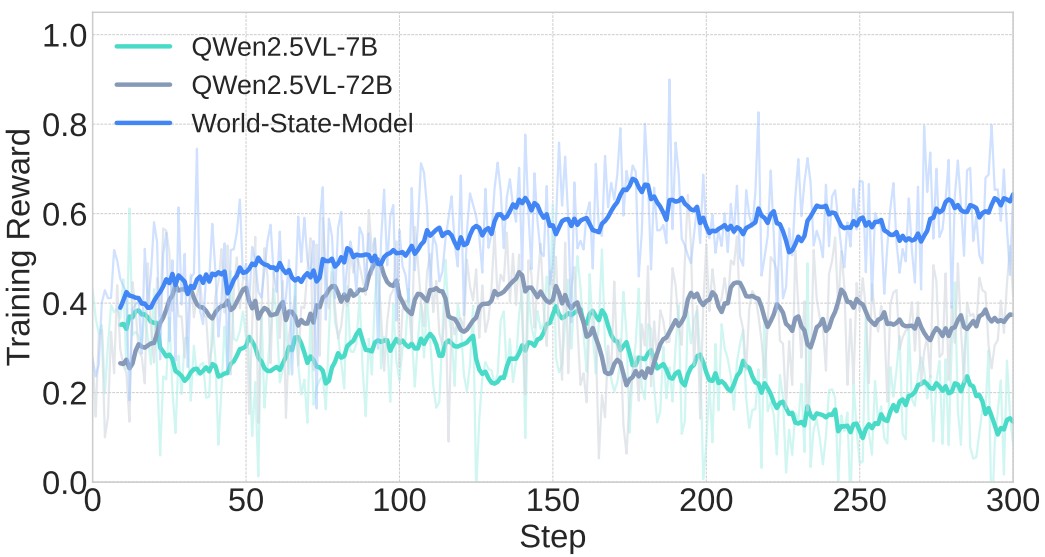

Figure 5: **Training reward with different reward signal provider.** Our World State Model provide reward signal that can achieve improved training reward compared to strong base models.

## A    WORLD STATE MODEL

The World State Model (WSM) is a central component of SEAgent, responsible for understanding visual state changes and evaluating the effectiveness of the agent's actions.

### A.1    MODEL ARCHITECTURE AND OPERATION

The WSM is built upon the Qwen2.5-VL-7B vision-language model. It operates in two distinct modes, each with a specific input-output structure to perform different tasks:

1. **Trajectory Judgment:**

   **Input:** A sequence of screenshot images captured during an episode.

   **Output:** Short captions for each screenshot, the reasoning process for the judgment, and a structured judgment dictionary (containing fields such as `Correctness`, `Redundant`, and `First Error Step`, as detailed in Fig. 8 of the supplementary material).

2. **State Change Description:**

   **Input:** Two screenshot images, one from before and one after a single action was executed.

   **Output:** A detailed description of the visual differences between the two images.

### A.2    FINE-TUNING DATASET AND PROCESS

To equip the WSM with these capabilities, a specialized dataset was constructed for fine-tuning.

**Data Construction**  The data construction process is as follows:

1. **Trajectory Sampling:** A Computer Using Agent (CUA), powered by UI-TARS and Gemini-2.5-Pro, was used to sample trajectories from 43 feasible tasks in Google Chrome within the OSWorld benchmark. These trajectories were saved as screenshot sequences.

2. **GPT-4o Annotation:** Using the prompts detailed in Figures 6 and 7 of the supplementary material, GPT-4o was employed to annotate the sampled trajectories, generating judgments and screenshot captions. Only samples where the judgment matched the ground truth from OSWorld evaluation protocols were retained, resulting in 860 high-quality annotated trajectories.

3. **Change Description Data:** An additional 1,000 pairs of (before action, after action) screenshots were sampled. GPT-4o was used to generate detailed descriptions of the differences, creating a 1,000-sample Change Description (CD) dataset.

**Fine-Tuning Process**  The fine-tuning was performed using the Llama-Factory framework on 8 NVIDIA A100 (80G) GPUs for 2,000 iterations. A learning rate of $2 \times 10^{-5}$ was used, and LoRA (rank=128) was employed for parameter-efficient fine-tuning. The 860 annotated trajectories serve as the core training data for teaching the model trajectory judgment, captioning, and reasoning. The 1,000-sample CD dataset acts as auxiliary data, specifically to encourage the model to focus on fine-grained visual differences, which enhances its overall state understanding. As shown in Table 1 of the main paper, incorporating CD data significantly boosts judgment performance. The two datasets were combined for training without any special re-weighting.

## A.3 Reward Generation from Trajectory Analysis

The trajectory judgment capability of the WSM is the core source of the reward signal for reinforcement learning. After an agent executes a full trajectory $\mathcal{H} = \{s_0, a_0, s_1, a_1, \ldots, s_{\text{final}}\}$, the WSM analyzes it and outputs a structured judgment. Based on this output, actions within the trajectory are dynamically labeled as either positive actions ($a_T$) or failure actions ($a_F$):

- **Fully Successful Trajectory:** If `Correctness` is 'True' and there are no `Redundant` steps, all actions $a$ in the trajectory are labeled as $a_T$.
- **Successful but Inefficient Trajectory:** If `Correctness` is 'True' but `Redundant` steps begin at step $k$, all actions prior to step $k$ are labeled as $a_T$.
- **Failed Trajectory:** If `Correctness` is 'False' and the `First Error Step` is $e$, all actions prior to step $e$ are labeled as $a_T$, while the erroneous action $a_e$ is labeled as $a_F$.

These dynamically labeled $a_T$ and $a_F$ actions constitute the reward signals for the RL pipeline. During training, the actor predicts an action $a_t$ based on the history $\{a_0, s_0, \ldots, s_t\}$ and uses these labels to calculate rewards.

# B  Curriculum Generator

The Curriculum Generator is designed to dynamically produce tasks of increasing difficulty and diversity, guiding the agent through a systematic exploration of the software's capabilities.

## B.1 Task Generation Mechanism

The workflow of the Curriculum Generator is detailed in the pseudocode in our supplementary material. Its core idea is to leverage the WSM's analysis of completed tasks to generate new ones. The process, illustrated by the "add a rectangle" example from Figure 5, involves three main steps:

1. **Analysis and Feedback:** The agent successfully completes an initial task, "add a rectangle." The WSM analyzes the execution trajectory and extracts two key pieces of information: a task evaluation (`Exam`) and a list of observed state changes (`CD_list`).

    `CD_list`: {"add a rectangle": ["The Edit bar is expanded...", "The cursor has changed into a cross...", "A blue box appears on the screen with side bars showing

properties such as fill, line, color, width, transparency, and corner style...”], ...}
Exam: [{”task”: ”add a rectangle”, ”status”: ”success”}, ...]

2. **Knowledge Integration and Task Generation:** The `CD_list` and `Exam` are fed into the Curriculum Generator. It distills new knowledge, such as ”properties of a rectangle,” and integrates it into its internal `Software guidebook`. Based on this new knowledge, it generates more challenging tasks like ”Add a green rectangle” or ”Add a red rectangle with 50% transparency,” which are then added to the task buffer.

3. **Iterative Learning:** In the next RL phase, the agent samples from this updated, more challenging task buffer. The continuously enriched `Software guidebook` acts as the system's long-term memory, driving the Curriculum Generator to propose increasingly sophisticated and unexplored tasks in subsequent rounds, thereby guiding the agent toward mastery.

## C    DETAILS OF CURRICULUM GENERATOR.

### C.1    EXEMPLAR CASE DURING TASK EVOLUTION.

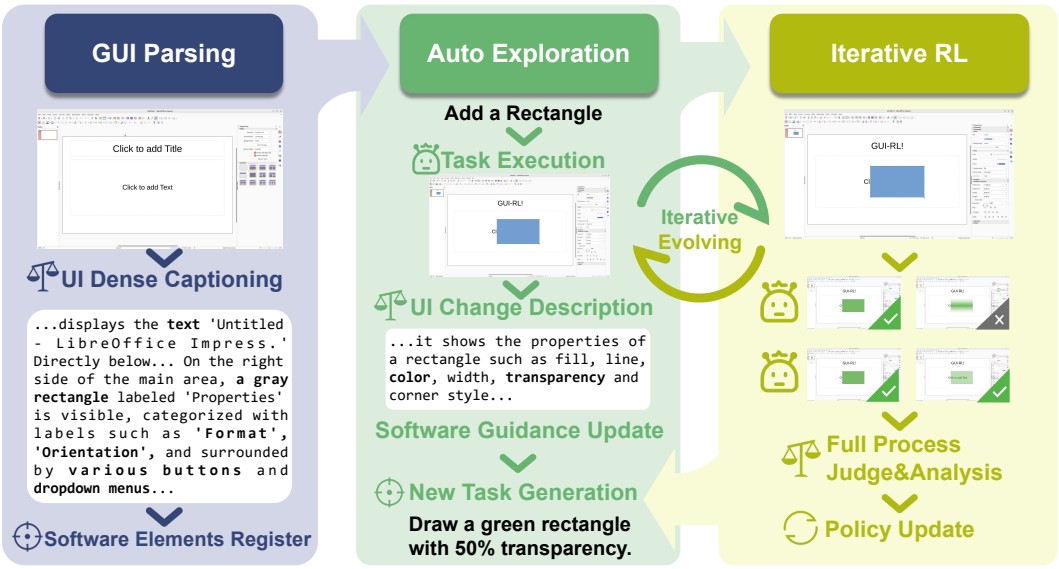

Figure 6: **SEAgent autonomous exploration pipeline.** The agent (policy model) and World State Model iteratively generate new task and perform RL to become a specialist in novel software.

We provide an exemplar case of our task evolution pipeline in Fig. 6, demonstrated using LibreOffice Impress. Initially, the World State Model parses a screenshot of the Impress interface into detailed captions describing the layout and individual buttons. The Task Generator then produces an initial task set, $\mathcal{I}_0 = \{I_0^{(1)}, I_0^{(2)}, \ldots\}$, and summarizes the initial software guidance memory $U_0$. The initial agent executes tasks in $\mathcal{I}_0$, such as "Add a Rectangle," while the World State Model evaluates these actions, providing judgments and detailed descriptions of resulting changes. As shown in the Auto-Exploration stage, this includes generating captions for newly appeared property panels and assessing execution success. The Task Generator incorporates feedback on execution success and newly revealed properties (e.g., transparency) to evolve new tasks, such as "Draw a green rectangle with 50% transparency." This process iteratively improves through reinforcement learning, enabling continuous task evolution and agent self-improvement.

### C.2    COMPARATIVE ANALYSIS OF INSTRUCTION GENERATION STRATEGIES.

To validate the effectiveness of our Curriculum Generator, we conducted a comparative analysis against state-of-the-art instruction generation methods, namely those from NNetNav (Murty et al., 2025) and WebRL (Qi et al., 2024).

**Experimental Setup** We adapted the official code and prompts from these prior works from web environments to general software applications. To ensure a fair comparison of the curriculum quality, for each strategy, we employed two leading LLMs: the open-source Qwen2.5-72B (Bai et al., 2025) and the proprietary Gemini-2.5-Pro (Google DeepMind, 2025). The tasks generated by each strategy were used to train an RL agent (using GRPO only), with reward signals uniformly provided by our fine-tuned WSM. The evaluation was performed on two applications: VSCode from OSWorld (a standard software) and Celestia from ScienceBoard (?) (a more challenging, out-of-domain scientific application). The primary metric was the task success rate.

Table 5: Success rate (%) comparison of different task generation strategies on two software applications.

| Task Generation Strategy | LLM | VSCode | Celestia |
|---|---|---|---|
| WebRL | Qwen2.5-72B | 27.5 | 0.00 |
| WebRL | Gemini2.5-Pro-thinking | 36.2 | 3.03 |
| NNetNav | Qwen2.5-72B | 34.6 | 0.00 |
| NNetNav | Gemini2.5-Pro-thinking | 43.6 | 5.05 |
| Curriculum Generator (Ours) | Qwen2.5-72B | 37.7 | 9.09 |
| Curriculum Generator (Ours) | Gemini2.5-Pro-thinking | 42.3 | 12.12 |

**Results and Discussion** The results are presented in Table 5. As shown, the reverse instruction generation strategy from NNetNav (Murty et al., 2025) is highly effective on the in-domain application (VSCode), demonstrating high data generation efficiency by producing successful trajectories. However, a critical trade-off was observed: this approach tends to generate many similar tasks, limiting its ability to explore the full breadth of the software's functionalities. This limitation becomes more pronounced when the task generator is unfamiliar with the target software, as seen in the OOD Celestia environment.

In contrast, our guidebook-based method, while having a lower initial data generation efficiency, excels at systematic exploration. It builds structured knowledge of the software from scratch, making it more robust for tackling novel applications. This is evidenced by its superior performance on the more challenging Celestia software.

We conclude that these two strategies are complementary. Reverse instruction generation can efficiently exploit known functionalities, while our guidebook-based method can systematically explore new ones and help the task generator build a more comprehensive understanding of the target software. A hybrid approach combining both strategies is a promising direction for future work.

## D  EXPERIMENTS ON ANDROIDWORLD

Table 6: Success Rate on AndroidWorld (Rawles et al., 2024)

| Model | AndroidWorld_SR |
|---|---|
| Qwen2.5-VL-7B | 8.0 |
| Qwen2.5-VL-7B+SEAgent | 19.5 |
| UI-TARS-7B-SFT | 33.0 |
| UI-TARS-7B-SFT+SEAgent | 38.0 |

To evaluate SEAgent's application to other format of GUI, we conduct new experiments on the AndroidWorld (Rawles et al., 2024) benchmark, which focuses on mobile GUIs. We apply our SEAgent pipeline to two distinct backbone models. As shown in the table below, our method yields substantial performance improvements for both, demonstrating that its self-evolving approach is effective across different model architectures and GUI formats. Specifically, SEAgent improves the success rate of Qwen2.5-VL by +11.5% and UI-TARS by +5.0%. This result strongly indicate the effectiveness of our pipeline also generalize to other form of GUI.

# E    SENSITIVITY ANALYSIS ON KEY HYPERPARAMETERS

We conducted a sensitivity analysis on key hyperparameters to evaluate their impact on the SEAgent pipeline. For model sampling, we set the temperature $t = 0$ for better reproducibility. We analyze two specific parameters: the number of generated tasks and the number of change descriptions. The results are presented in Table 7 and discussed below.

Table 7: Sensitivity analysis for key hyperparameters in the SEAgent pipeline, evaluated on VSCode. The metric is Success Rate (%).

| # Tasks Generated | VScode SR | # Change Descriptions | VScode SR |
|---|---|---|---|
| 30 | 31.88 | 30 | 33.33 |
| 50 | 36.23 | 50 | 37.68 |
| 100 | 37.68 | 100 | 37.68 |
| 200 | 37.68 | 200 | 34.78 |

**Number of Generated Tasks**    This parameter controls the breadth of exploration in each learning cycle. As shown in our analysis, performance improves as more diverse tasks are generated, eventually plateauing around 100 tasks.

**Number of Change Descriptions**    This parameter controls how much new information the generator receives to update its "software guidebook." We found a clear trade-off: A sufficient number of descriptions (50–100) is essential for the generator to learn about new UI functionalities and create meaningful, unexplored tasks. However, providing too many descriptions (e.g., 200) creates an overly long context for the LLM, which degrades the quality of task generation and hurts final performance.

# F    ABLATION ON THE LOSS BALANCE FACTOR.

In Sec.3.2, we use $\gamma$ to balance the ratio of two loss item: adversarial imitation that learn from error and GRPO that learn to achieve success. We ablate the choice of $\gamma$ in Tab.8, according to which we set $\gamma = 0.2$ in main experiments.

| $\gamma$ | 0.0 | 0.1 | 0.2 | 0.3 | 0.5 | 0.8 |
|---|---|---|---|---|---|---|
| Success Rate (%) | 34.8 | 36.2 | 37.7 | 31.9 | 26.1 | 23.1 |

Table 8: VScode Success Rate on OSWorld (Xie et al., 2024) under different loss balance factor $\gamma$ values.

# G    REWARD FUNCTION FOR DIFFERENT ACTIONS.

| Action Type | Description | Distance-based Reward |
|---|---|---|
| click, left_single, right_single, hover | Click or hover on a location | Normalized L1 distance between predicted and ground-truth coordinates |
| left_double, double_click | Double click on a region | Normalized L1 distance between clicked coordinates |
| drag, select | Drag from start box to end box | Intersection over Union (IoU) between predicted and ground-truth boxes |
| type | Type textual input | Character-level BLEU score between predicted and ground-truth text |
| hotkey | Press multiple keys at once | Character-level BLEU score between predicted and ground-truth key combinations |
| press | Press a single key | Character-level BLEU score between predicted and ground-truth key |
| scroll | Scroll in a certain direction | Character-level BLEU score between predicted and ground-truth direction |
| move_mouse | Move mouse to a specific location | Normalized L1 distance between predicted and ground-truth coordinates |
| highlight | Highlight a rectangular UI region | IoU between predicted and ground-truth region |
| copy, paste | Clipboard operations | BLEU score between copied/pasted content |
| wait | Explicit wait command | Fixed reward + 1 |
| finished, finish_task | Finish current task/trajectory | Fixed reward + 1 |

Table 9: Reward computation for each action type in GUI agent

|          | Phase0  | Phase1  | Phase2  | Phase3 |
|----------|---------|---------|---------|--------|
| VSCode   | 112/39  | 282/83  | 161/34  | 98/55  |
| GIMP     | 104/51  | 309/90  | 183/50  | 95/52  |
| Impress  | 102/44  | 290/92  | 185/61  | 87/51  |
| VLC      | 85/29   | 114/41  | 160/48  | 53/27  |
| Writer   | 123/62  | 278/101 | 201/69  | 101/43 |

Table 10: Number of episode (Success/Failure) across four phases for different software tools during self-evolution. Each episode contains 8.8 multi-turn conversions in average.

## H    DATA STATISTICS DURING ITERATIVE REINFORCEMENT LEARNING.

## I    DETAILED PROMPT TEMPLATES.

For evaluation on AgentRewardBench (Lù et al., 2025), we use their official template for final state screenshot only testing and modified prompt in Fig.7 for entire process (or sampled middle screenshots) testing.

For evaluation on OSWorld Sampled trajectories, we use prompt in Fig.8 to prompt GPT-4o to provide step level judges, the sampled judges on Chrome in OSWorld (Xie et al., 2024) serves as training data of GUI-Judge. This template is also used in training GUI-Judge and at inference time in autonomous exploration stage.

For navigator, we use prompt template in Fig.9, which takes previous software usage manual and the performance of actor agent evaluated by judge (Empty if in initial phase.) as well as detailed exploration caption as input and output the updated usage manual as well as new task for agent to execute.

## J    SELF DOCUMENTED USAGE MANUAL ON DIFFERENT SOFTWARE DURING EXPLORATION.

In Fig.10 Fig.12, Fig.11, Fig.13, we demonstrate the self-documented usage manuals of the navigator (Qwen2.5-72B (Yang et al., 2024)) in the exploration and learning system introduced in Sec.3.1.

## K    BROADER IMPACTS

**Potential positive societal impacts:** SEAgent introduces a self-evolving paradigm for Computer Use Agents (CUAs), enabling them to autonomously learn and adapt to previously unseen software without human supervision. This significantly reduces the need for extensive manual data annotation and domain-specific customization, allowing intelligent agents to assist users across a wide range of applications—including productivity tools, multimedia editing, and educational software. By automating repetitive tasks and providing guidance in complex software environments, SEAgent holds promise for improving accessibility, enhancing digital literacy, and reducing cognitive workload in both professional and everyday settings.

**Potential negative societal impacts:** The capability of SEAgent to autonomously explore and operate complex software also introduces risks of misuse. Malicious actors might repurpose SEAgent for unauthorized software automation, such as automating account creation, spamming interfaces, or conducting surveillance via GUI interactions. In addition, as the agent learns from its own experience, there exists a risk that the agent may inadvertently inherit or amplify software-specific biases, potentially leading to unfair or inappropriate behaviors in sensitive applications (e.g., finance, legal automation). Mitigation strategies include controlled release of models, behavior filters during deployment, and incorporating safeguards in the World State Model to detect and prevent unintended or adversarial behavior.

---

**Algorithm 1** SEAgent Specialized Self-Evolution Training Loop

---

1: **Input:** Initial policy $\pi_0$, World State Model $\mathcal{M}_{\text{state}}$, Curriculum Generator $\mathcal{M}_{\text{task}}$, Initial GUI state $S_0$
2: **1. Task Initialization**
3: $\mathcal{C}_0 \leftarrow \text{CaptionGUI}(S_0)$          ▷ Parse initial GUI layout (menu bar, buttons, etc.)
4: $\mathcal{I}_0, U_0 \leftarrow \mathcal{M}_{\text{task}}(\emptyset, \emptyset, \emptyset, \mathcal{C}_0)$          ▷ Generate basic initial tasks and usage guide

---

5: **for** $p = 0$ to $P - 1$ **do**          ▷ 2. Self-Evolution Phase Loop
6:      **2.1 Autonomous Exploration**
7:      $\mathcal{D}_{\text{traj}} \leftarrow \emptyset$
8:      **for all** $I \in \mathcal{I}_p$ **do**
9:          $\tau \leftarrow \text{ExecuteInstruction}(\pi_p, I)$      ▷ Actor executes task in the virtual environment
10:          **2.2 Effect Evaluation**
11:          $\mathcal{J}_I, \mathcal{C}_I \leftarrow \mathcal{M}_{\text{state}}(\tau)$      ▷ Step-level trajectory judgment and new state captions
12:          $\mathcal{D}_{\text{traj}} \leftarrow \mathcal{D}_{\text{traj}} \cup \{(\tau, \mathcal{J}_I, \mathcal{C}_I)\}$    ▷ $\mathcal{J}_I$: a sequence of per-step feedback labels ($a_T$ or $a_F$)
13:      **end for**

---

14:      **2.3 Policy Update (RFT)**
15:      Split $\mathcal{D}_{\text{traj}}$ into:
16:          $\mathcal{D}_{\text{pos}}$: steps labeled as positive $a_T$
17:          $\mathcal{D}_{\text{neg}}$: steps labeled as negative $a_F$
18:      Compute GRPO loss on $\mathcal{D}_{\text{pos}}$:
19:      $r(a, a_T) = \mathbb{I}[\text{type}(a) = \text{type}(a_T)] + r_{\text{dist}}(a, a_T)$
20:      Compute Adversarial Imitation loss on $\mathcal{D}_{\text{neg}}$:
21:      $\mathcal{L}_{\text{AI}} = -\log \frac{\pi_\theta(a|s,I)}{\pi_{\text{ref}}(a_F|s,I)}$
22:      Total loss: $\mathcal{L}_{\text{total}} = \mathcal{L}_{\text{GRPO}} + \gamma \mathcal{L}_{\text{AI}}$
23:      $\pi_{p+1} \leftarrow \text{Update}(\pi_p, \mathcal{L}_{\text{total}})$

---

24:      **2.4 Task Update**
25:      $\mathcal{I}_{p+1}, U_{p+1} \leftarrow \mathcal{M}_{\text{task}}(U_p, \mathcal{I}_p, \{\mathcal{J}_I\}, \{\mathcal{C}_I\})$    ▷ Generate more complex tasks based on new software knowledge and performance feedback
26: **end for**

---

27: **Output:** Specialized agent policy $\pi_P$ after $P$ stages of self-evolution

---

> **Web Step Level Judge Prompt Template**
>
> You are a Language Model specialized in judging the performance of web agents in web-navigation tasks. For a certain website, you are given the goal of a navigation task, the current URL of the webpage, the actions taken by the agent, and the thought process of the agent. **Additionally, you will have access to the sequence of key frame screenshots** Your task is to answer several questions about the agent's performance in the task.
> **You should carefully look at the sequential screenshot images in order to decide whether its sucessfully finish the task or failed halfway.**
>
> Question 1: Was the sequence of actions successful in achieving the goal?
> Choices: <success>Successful</success>, <success>Unsuccessful</success>
>
> Question 2: Did the agent perform unnecessary actions that could lead to unintended side effects?
> Choices: <side>Yes</side>, <side>No</side>
>
> Question 3: Did the agent perform the task optimally, by only performing necessary actions and avoiding unnecessary ones?
> Choices:
> <optimal>1. Complete Failure</optimal>
> <optimal>2. Suboptimal</optimal>
> <optimal>3. Somewhat Optimal</optimal>
> <optimal>4. Completely Optimal</optimal>
>
> Question 4: Did the agent loop through a sequence of actions that did not make progress towards the goal?
> Choices: <loop>Yes</loop>, <loop>No</loop>
>
> Provide your reasoning for each question.
> Your answer **must** follow this exact format:
>
> <reasoning>your reasoning here</reasoning>
> <success>answer</success>
> <side>answer</side>
> <optimal>answer</optimal>
> <loop>answer</loop>

Figure 7: **Prompt Template of GUI-Judge for web agent trajectories evaluations** with history screenshots as input, its difference with default prompt of AgentRewardBench (Lù et al., 2025) is highlighted in bold.

## L  SEAGENT SELF-EVOLUTION ALGORITHM

Algorithm 1 presents the core self-evolution training loop of SEAgent in a specialized software environment. The procedure is divided into four major stages:

(1) **Task Initialization.** Given the initial GUI state of a target software application, the World State Model performs dense captioning to extract structural semantics (e.g., menu bar, buttons), which is used by the Curriculum Generator to create an initial set of executable tasks and an editable software guidebook.

(2) **Autonomous Exploration and Effect Evaluation.** The agent explores each task via its current policy. The World State Model then performs step-level trajectory analysis, assigning each action a feedback label—either correct ($a_T$) or incorrect ($a_F$)—and generating GUI state change captions. This produces rich supervision signals for both policy learning and downstream task generation.

(3) **Policy Update via Reinforcement Fine-Tuning.** Based on the labeled execution data, positive and negative action steps are separated. We apply Group Relative Policy Optimization (GRPO) to reinforce correct actions, and Adversarial Imitation (AI) to suppress failure-prone behaviors. The updated policy is used for the next exploration round.

---

### OSWorld Step Level Judge Prompt Template

I am evaluating the performance of a UI agent. The images provided are sequential keyframes that represent the full execution trajectory of the agent when attempting to follow a command. These keyframes correspond to the instruction: [INSTRUCTION].

Please thoroughly analyze the sequence to assess the following aspects:

1. Correctness — Did the agent successfully complete the task as instructed?
2. Redundant Steps — Identify any unnecessary or repeated actions that do not contribute to the goal.
3. Optimization — Did the agent follow an efficient plan with a minimal number of steps?
5. First Error Step — If the execution is incorrect or sub-optimal, determine the index of the first 5. keyframe where a mistake occurred.
6. Error Analysis — Provide a brief explanation of the mistake at that step.
7. Correct Action Suggestion — Explain what the agent should have done instead at the point of error.

Important Instructions:
The agent may have made progress toward the goal, but unless the task is fully and correctly completed, you must set 'Correctness' to False.
Be cautious in determining success. Missing confirmation screens, skipped inputs, or wrong UI elements clicked all count as errors.
Carefully examine all UI changes, button interactions, text entries, and any visual feedback in the screenshots.
Clearly indicate which exact steps are redundant (starting from 1).
Once you finish the analysis, return your evaluation in the following dictionary format. Include your step-by-step reasoning above the result.

```
<thinking>step by step reasoning.</thinking>
res_dict = {
    "Correctness": True or False,
    "Redundant": [step numbers],
    "Optimized": True or False,
    "First_Error_Step": step number or None,
    "Error_Type": "brief description of the mistake",
    "Correct_Action": "what should have been done instead"
}
```

Figure 8: **Prompt Template of GUI-Judge for OSWorld (Xie et al., 2024) trajectories**, which prompts judge model to provide step level reward signal.

(4) **Task Update.** The Curriculum Generator leverages feedback signals ($\mathcal{J}$) and GUI state transitions ($\mathcal{C}$) to propose more diverse and challenging tasks, thereby expanding the task frontier in a curriculum fashion.

This process repeats over multiple curriculum phases, ultimately yielding a specialized agent policy capable of mastering complex operations in the given software environment.

---

### Task Buffer Update Prompt Template

You are now a teacher training a Computer Use Agent (CUA). This CUA is exposed to a new software environment and undergoes multiple rounds of iterative training. Your task is to issue new tasks for the agent to explore and train on, based on the feedback from the agent's actions. You are also responsible for summarizing a software usage manual to help the agent remember knowledge about the software.

The agent has provided the following feedback on its operations within the software: {json.dumps(action_decription_list)}

Here is the software usage document you summarized in the previous round: {document}

Here is the agent's performance on the task you provided in the previous round: {json.dumps(exam)}

Your are also access to the previous given tasks with the screenshot caption after agent's execution. You can also use these captions and results to evaluate the agent's capability and generate new task and update document accordingly given the caption of the new screen and the corresponding intruction with judged evaluation: {json.dumps(prev_states)}

Please:
- Analyze the agent's performance.
- Integrate new knowledge from the feedback.
- Update the usage manual accordingly.
- Design a new set of tasks (with increased difficulty) (30 or more) that reinforce the concepts the agent struggled with in the last round.
- Each task **must be concise and specific**, targeting a concrete atomic action, based on the document and agent's observations, such as:
    - "Create a file named main.py."
    - "Open Terminal card."
- Each task must be executable from software initial state with no file open, e.g. you should not generate task like save xxx.txt if xxx.txt doesn't exist or created.
- if task is in sequencial order with reliance, you should output a seq list like [subtask1, subtask2, ...], if there is no reliance, output [task].
- Decompose and target previous errors in a more focused way.

Output your reasoning and analysis process first. Then output the updated usage document and task list in the following JSON format within a SINGLE JSON DICT easier for me to parse:

```json
{{
    "software_document_new": "...",
    "exam_new": [[subtask1, subtask2, ...], [task]...]
}}
```

Figure 9: **Prompt Template for task buffer update**, which generates new tasks in a curriculum manner and update software documents. The new tasks are used for actor to perform next phase of RL.

Figure 10: **Automatically generated usage manual during self exploration** on VScode.

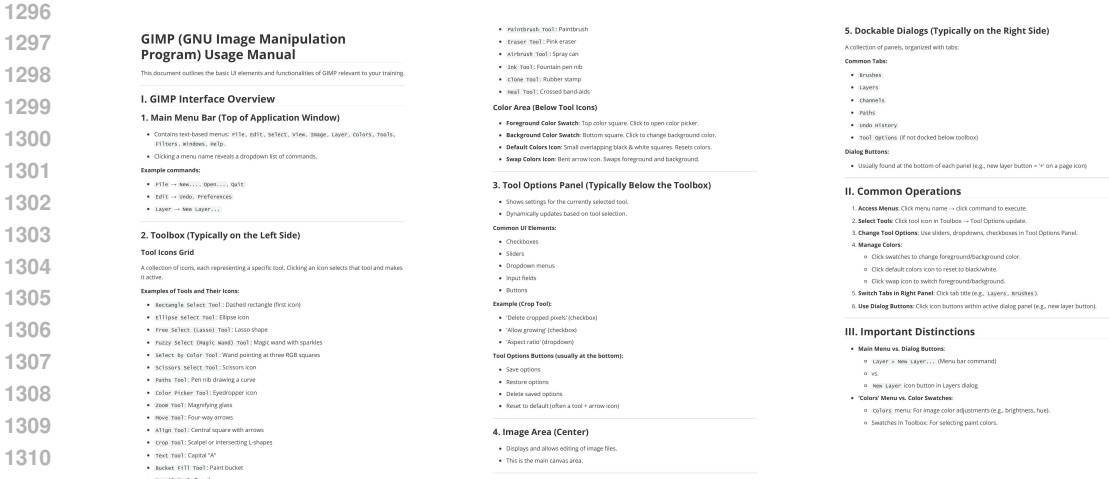

Figure 11: **Automatically generated usage manual during self exploration** on GIMP.

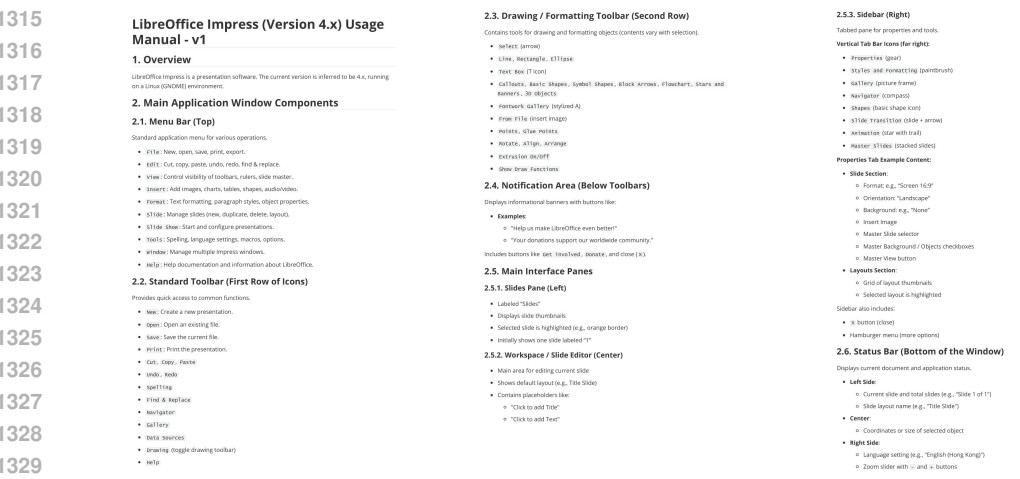

Figure 12: **Automatically generated usage manual during self exploration** on LibreOffice_Impress.

Figure 13: **Automatically generated usage manual during self exploration** on LibreOffice_Writer.

