# OpenReview forum: "SEAgent: Self-Evolving Computer Use Agent with Autonomous Learning from Experience"
_ICLR.cc/2026/Conference — ICLR 2026 Conference Withdrawn Submission_

### Official Review · Reviewer_PLT9 · 2025-10-26

**Soundness:** 2
**Presentation:** 1
**Contribution:** 3
**Rating:** 4
**Confidence:** 4

**Summary:**

The paper proposes an agentic self-evloving pipeline for learning autonomous computer use agent in software environments without human-labeled data.

The pipeline consists of several key components: (1) a World State Model to provide step-level reward signals for RL training and state change captioning as well as the trajectory-level judgement for maintaining a software guidebook for a Curriculum Generator; (2) a Curriculum Generator that automatically generates new task instructions for curriculum learning of the agent; (3) a training process involving adversarial imitation of failure actions and GRPO based on successful actions; (4) a specialist-to-generalist training strategy that  involves SFT on successful trajectories from specialist agents first and then RL across multiple softwares.

Experiments on OSWorld, ScienceBoard and AndroidWorld show the effectiveness of proposed pipeline.

**Strengths:**

1. This work identifies several key elements contributing to the development of autonumous agents for computer use without relying on annotated data, providing insights for practical implementation of such agents. The key elements include:

(1) a new task generation strategy using a Curriculum Generator;

(2) a new reward model to assess both the step-level actions and trajectory-level success using a World State Model;

(3) a new training objective combining both adversarial imitation of failure actions and RL (using GRPO).

(4) a specialist to generalist strategy that outperforms both direct RL in multiple softwares and RL in each environment.

2. Extensive experiments are conducted to justify different aspects, including the effectiveness on different benchmarks and using different actor models, the performance of World State Model compared to other models as judges, the comparison between Specialist RL, General RL and Specialist-to-Generalist, the performance gains during different training phases and ablation studies to show the usefulness of different components in the poposed pipeline.

**Weaknesses:**

1. The writing of the paper is too poorly, requiring many efforts to revise and polish.

(1) Too many grammatical errors, making it hard to read.

(2) Captions of most Tables and Figures are unclear, lacking explanations to details in the Table or Figure. For example, the meaning of different colors of the curves in Figure 4.

(3) The experimental setup is missing in the main paper, including benchmarks, baselines, evaluation metrics, and implementation details, making it hard to make a fair comparison between methods (although I can find some in the appendix).

(4) Some notions or details are not explained. For example, the explanation of *t* in Eq. 5, and how to construct the training data for GRPO? For other issues, please see questions below.

2. The introduction and comparison with related work is insufficient.

(1) The paper claims that "SEAgent achieves superior performance, which can be attributed to World State Model providing fine-grained, step-level rewards from the full history". However, similar ideas has been justified in [1] but not referred to, where a process reward model is prompted by the full interaction history.

(2) Some related work on GUI agents are missing and not discussed, e,g., [2][3]

3. It seems that the training sample for GRPO is a one-step data from a trajectory instead of a whole trajectory. Why use the single-turn RL in an agentic setting? What if the proposed training method is compared to multi-turn RL approaches?

References:

[1] [EPO: Explicit Policy Optimization for Strategic Reasoning in LLMs via Reinforcement Learning](https://aclanthology.org/2025.acl-long.747/) (Liu et al., ACL 2025)

[2] Luo, Run, et al. "Gui-r1: A generalist r1-style vision-language action model for gui agents." arXiv preprint arXiv:2504.10458 (2025).

[3] Chen, Kevin, et al. "Reinforcement learning for long-horizon interactive llm agents." arXiv preprint arXiv:2502.01600 (2025).

**Questions:**

1. Why do performance gains saturate beyond the thrid training phase?
2. What do the ensembled results of specialized RL mean in Table 2? How to understand the "individual specialist ensemble" in line 429?
3. Does **General RL** refer to training a single agent in all five environments simultaneously? If so, how to train a single agent in different environments through RL? And it seems that the only difference between **General RL** and **Specialist-to-Generalist** is that the latter involves SFT initialization before RL.
4. In all the experimental results, does SEAgent refer to SEAgent (Specialist-to-Generalist) by default?
5. There is an severe error in the paragraph of **Ablation Study of Specialist Training**, of which the content is not the analysis of ablation results. Also, how to understand Table 4? For example, Does Qwen2.5VL-72B refer to Curriculum Generator? What does the result in the first row mean?
6. The result in the last row of Table 4 is 46.1, which equals to that of SEAgent (Specialized RL) in Table 2. What are the differences between the two experiments in terms of its configuration?

---

> ### Author Response · Authors · 2025-11-13
>
> ### W1: Writing and Presentation
> We will dedicate significant effort to revising and polishing the entire manuscript.
> * **W1-1 (Grammar):** We will perform a thorough copy-edit to correct all grammatical errors and improve readability.
> * **W1-2 (Captions):** We will revise all Table and Figure captions to be self-contained and clearly explain all elements (e.g., the meaning of colors in Figure 4).
> * **W1-3 (Experimental Setup):** We used standard, fully open-source benchmarks (OSWorld, etc.) and their established evaluation methods. We will add a clear description of the benchmarks, baselines, and metrics to the main paper.
> * **W1-4 (Notions & Details):** '$t$' in Eq. 5 is the standard notation for the action timestamp (step number) within a trajectory.  The data construction for GRPO is detailed in Appendix A and B, and we will add a clearer reference to this in the main text.
> ***
>
> ### W2: Related Work
> Thank you for pointing out these important references. We will add a detailed discussion comparing our work to [1] (EPO), [2] (Gui-r1), and [3] (RL for long-horizon agents) in our revised related work section.
> ***
> ### W3: Single-turn vs. Multi-turn RL
> We view our GRPO implementation as a "Reinforcement Fine Tuning" (RFT) process. In our exploration, we found that the primary challenge of multi-turn RL in this domain is the sparse reward signal—a single late-stage error can invalidate an entire trajectory, causing correct early steps to be penalized and making learning highly inefficient.
> We have already compared our method against multi-turn RL baselines (WebRL and DigiRL) using PPO and demonstrated superior performance. While we believe an end-to-end multi-turn GRPO is a valuable direction for future work, our current method provides a more stable and effective solution. We will add this discussion to the revision.
> ***
>
> ### Q1: Performance Saturation
>
> Our current experimental setup limits all tasks to a 15-step horizon. The SOTA closed-source model (Claude 4.5) achieves success rate of 47% under the same 15-step constraint. We believe our S2G result of 40% is approaching the saturation point *for this specific task setting*.
> ***
> ### Q2: Ensembled Results of Specialized RL
> The "individual specialist ensemble" s not a single model. It represents the collective performance of the five specialist agents. The success rate is achieved by testing each specialist agent only on its corresponding software.
> ***
> ### Q3: General RL vs. Specialist-to-Generalist
> "General RL" refers to training a single agent across all five software environments simultaneously (which is possible as they all run on the same Linux system). The primary difference,  that our S2G method initializes the agent with SFT from the five specialist models *before* starting general RL. Our results show this is crucial. Specialized RL (focusing on one software) creates highly effective experts, and the SFT phase allows the generalist model to "inherit" these diverse capabilities, which is far more effective than training a generalist from scratch.
> ***
> ### Q4: Default "SEAgent" terminology
> Thank you for the clarification question. Yes, you are correct. "SEAgent" stands for Self-Evolving Agent. In the experimental results, "SEAgent" used without a specifier **refers to the performance of our full proposed methodology**, which includes both the World State Model (WSM) and the Specialist-to-Generalist (S2G) strategy.
> ***
>
> ### Q5: Understanding Table 4
> We apologize for the confusion regarding Table 4.
> * **First Row (Qwen2.5VL-72B):** This row represents an ablation where we replaced our trained World State Model (WSM) with the **base Qwen2.5VL-72B model as the judge** to provide reward signals. The significantly lower performance demonstrates that the base model is insufficient for this task and validates the necessity and importance of our specialized WSM in providing accurate, high-quality reward signals. The other rows (e.g., 'w/o AI', 'GRPO vs. SFT') are similarly ablations to validate the specific contributions of each component of our framework.
> ***
>
> ### Q6: Table 4 (Last Row) vs. Table 2 (Specialized RL)
> The last row of Table 4 ("w/o S2G," 46.1) is an ablation study on **Specialist training**. Its configuration is therefore **identical** to the "SEAgent (Specialized RL)" experiment reported in Table 2 (46.1), which is why the results are the same. The primary ablation study for the S2G strategy is represented in Table 2 (i.e., S2G vs. General RL vs. Specialized RL).
> ***
> We sincerely thank you for your valuable and constructive feedback. We will integrate these insightful comments into our next revision. Unfortunately, the current rating puts us in a difficult position. We are approaching a deadline for another suitable conference. We sincerely hope our responses are sufficient to address your concerns. We would be very grateful for your timely re-evaluation, as we would strongly prefer to continue with this submission.

---

### Official Review · Reviewer_e9MR · 2025-10-29

**Soundness:** 4
**Presentation:** 4
**Contribution:** 4
**Rating:** 2
**Confidence:** 4

**Summary:**

This paper introduces SEAgent, an agentic self-evolving framework designed to enhance the adaptability of Computer Use Agents (CUAs) operating in previously unfamiliar software environments. The framework enables agents to engage in autonomous exploration and experiential learning, supported by a world state model and a curriculum generator. SEAgent optimizes its policy by using adversarial imitation and GRPO methods. With a specialist-to-generalist training strategy, SEAgent shows significant performance improvements across five professional software applications from OSWorld.

**Strengths:**

### 1. **Innovation**
   - **Self-evolving framework**:  **SEAgent**, a **self-evolving framework** for autonomous exploration and experiential learning, is innovative in CUAs fields. It allows agents to autonomously generate tasks and assess their success/failure in previously unfamiliar software environments without human intervention, advancing the capabilities of autonomous systems.

### 2. **Task Generation and Evaluation Precision**
   - **World State Model**: The paper introduces the **World State Model**, which enables precise environmental state captioning and step-wise trajectory assessment. This model provides fine-grained reward signals, significantly improving the precision of task evaluation.

### 3. **Experimental Results and Performance Improvements**
   - **Multi-software environment adaptability**: SEAgent demonstrates excellent adaptability, not only excelling in single-software environments but also in **multi-software scenarios**. Its validation across five professional software applications from **OSWorld** showcases its generalization.

---

**Weaknesses:**

1. **Inconsistency in the Paper's Claims**
   The paper's claims are not fully self-consistent. Although the paper's title suggests **self-evolving** agents and emphasizes the exploration of LVLMs for autonomous exploration, the **World State Model** used for exploration in this study still relies on **human-annotated high-quality datasets**. This contradicts the starting point outlined in **line 53**, which advocates for agents to evolve without such dependencies. The paper should clarify this contradiction between the claimed self-evolving nature of the agent and its reliance on human-annotated data.

2. **Lack of Novelty in Contributions**
   The contributions of the paper lack sufficient innovation. The **self-taught**[1] research has already demonstrated that additional training using high-quality critic modules can guide the improvement of LLMs through **Supervised Fine-Tuning (SFT)**, but with limited gains from multiple rounds of training. The paper uses **Group Relative Policy Optimization (GRPO)**, a reinforcement learning method, for enhancement, but no additional conclusions are provided. One important question that remains unaddressed is whether more rounds of interaction could yield better results, as reinforcement learning theory suggests that continued interaction can lead to further improvements.
   On the other hand, the **specialist-to-generalist distillation method** may not be necessary in this context. In traditional RL, such a method is useful due to the small size of the network, where **Catastrophic Forgetting** may occur. However, for large models, multi-domain data mixing can be achieved during pretraining, and there is no clear need for separate training followed by distillation during the fine-tuning phase. The paper lacks ablation studies to analyze this point of innovation.

3. **Experimental Setup Issues**
   The experimental setup is not entirely reasonable.
   - **Table 1** does not provide results for the World State Model under the **LS setting**, leaving this aspect of the model's performance unexplored.
   - **Table 2** raises a concern: is the training data size consistent across **general RL**, **SFT**, and **specialist-to-generalist** methods? From the descriptions in the paper, it seems that the **specialist-to-generalist** approach would likely involve more training data, but this is not explicitly clarified.
   - **Table 4**: The ablation study does not fully incorporate the innovative aspects discussed in **point 2**, specifically regarding the necessity and impact of the specialist-to-generalist strategy.

4. **Writing Problems**
   The writing lacks rigor. For example, in **line 131**, the phrase should be "Recently, ... has achieved". Similarly, in **line 235**  "in Section [???]". Authors should check their writing more carefully.

[1] Wang, Tianlu, Ilia Kulikov, Olga Golovneva, Ping Yu, Weizhe Yuan, Jane Dwivedi-Yu, Richard Yuanzhe Pang, Maryam Fazel-Zarandi, Jason Weston, and Xian Li. "Self-taught evaluators." arXiv preprint arXiv:2408.02666 (2024).

**Questions:**

See weekness above.

---

> ### Author Response · Authors · 2025-11-13
>
> We thank you for your comprehensive review and insightful feedback on our paper, "SEAgent: A Self-Evolving Framework for Mastering Novel Software." We are very grateful for your recognition of our work and for highlighting its strengths.
> Below, we address each of your concerns in detail.
> ***
> ### W1: Inconsistency in the Paper's Claims
>
> We respectfully disagree with this assessment, as our main goal is to improve the model's judgment ability over long-sequence GUI trajectories (i.e., whether a task is completed).
>
> The term "self-evolving" refers to the agent's autonomous learning process (curriculum generation, exploration, and experiential learning) without human intervention in the loop. The WSM, as a foundational component, naturally requires high-quality training data, just as any base model does. For instance, the post-training of Qwen2.5-VL also involves substantial "human-annotated high-quality datasets," including GUI data. By this logic, any framework using a pre-trained model could not be called self-evolving.
>
> Furthermore, we have already verified that stronger base models, like Qwen3-VL, possess superior decision-making capabilities that directly translate to enhanced performance in our framework (e.g., boosting the VSCode benchmark success rate from 30.1% to 43.5%). This confirms our WSM architecture is sound and benefits from model advancements.
>
> Therefore, we respectfully maintain that our framework's "self-evolving" designation is appropriate, as its autonomous learning loop is independent of the initial, one-time data annotation required for the foundational model.
> ***
> ### W3-1: Experimental Setup (Missing LS results in Table 1)
>
> Thank you for pointing this out. The LS (last screenshot) results for our World State Model are indeed present in Table 1, and we apologize if they were not clear.
>
> Here are the specific LS results for our model from the table, which we will highlight more clearly in our revision:
>
> World State Model (w/o CD) [LS]:
>
> AgentRewardBenc: Prec 66.8, NPV 90.6
>
> OS-World-Full: Prec 52.5, NPV 85.8
>
> Prof/Office: Prec 57.1, NPV 81.1
>
> World State Model (w/ CD) [LS]:
>
> AgentRewardBenc: Prec 65.2, NPV 79.1
>
> OS-World-Full: Prec 49.3, NPV 82.0
>
> Prof/Office: Prec 55.8, NPV 77.2
> ***
> ### W3-2: Experimental Setup (Data size inconsistency)
>
> We acknowledge the concern regarding fair comparison. To ensure the training costs were consistent, we conducted an additional round of RL for the "general RL" method (which does not use our S2G strategy). This baseline was trained for the same number of RL steps and on the same amount of data (3.5k trajectories) as our S2G method. We will explicitly emphasize this in the revised manuscript to make the fairness of the comparison clear.
> ***
> ### W4: Writing Problems
>
> Thank you for catching these errors. We will thoroughly proofread the entire manuscript and correct all typographical and grammatical issues, including the ones you highlighted.
> ***
> We sincerely thank you for your valuable and constructive feedback. We will integrate these insightful comments into our next revision. Unfortunately, the current rating puts us in a difficult position. We are approaching a deadline for another suitable conference, which is forcing us to make an imminent decision about this submission. We sincerely hope our responses are sufficient to address your concerns. We would be very grateful for your timely re-evaluation, as we would strongly prefer to continue with this submission.

---

### Official Review · Reviewer_angx · 2025-10-30

**Soundness:** 3
**Presentation:** 2
**Contribution:** 3
**Rating:** 6
**Confidence:** 2

**Summary:**

To address the issue that CUAs often struggle with novel and specialized software, particularly in scenarios lacking human annotations, the authors propose SEAgent, an agentic self-evolving framework enabling CUAs to autonomously evolve through interactions with unfamiliar software. In addition, they introduce an effective specialist-to-generalist strategy for shaping a versatile generalist agent. The authors validate the effectiveness of their method across five professional software of OSWorld, ScienceBoard, and AndroidWorld.

**Strengths:**

- The lower part of Figure 1 presents the specialist-to-generalist training strategy with great clarity, and this strategy also provides valuable inspiration for progress in other domains. In general, each domain requires models to possess multi-dimensional capabilities, and the visualization and the demonstrated effectiveness of the proposed strategy in addressing this issue are particularly insightful and valuable.
- The paper provides a very clear introduction to the background of the task. Although I had not previously studied this field, I was able to gain a basic understanding of the task through reading the paper.
- Overall, the paper presents a carefully specific design of RL (such as reward signal), and the experimental section demonstrates a high degree of completeness.

**Weaknesses:**

- The difference between the specialist-to-generalist training strategy proposed in this paper and previous similar strategies requires further clarification.
- The readability of Section 3.1 and Figure 2 is not very good, and Figure 2 is not cited anywhere in the paper.
- There is a citation error at line 235.

**Questions:**

- Could the authors illustrate the workflow in Section 3 with a concrete example—specifically, how the evolution process unfolds and how each component functions?
- Is the term 'world state world' something you proposed? Or does the concept of world models also exist here?

---

> ### Author Response · Authors · 2025-11-13
>
> We thank you for your comprehensive review and insightful feedback on our paper, "SEAgent: A Self-Evolving Framework for Mastering Novel Software." We are very grateful for your recognition of our work and for highlighting its strengths. We will try to address your concerns and questions below.
> ***
> ### W1:
> Thank you for this question. Our key distinction lies in applying the specialist-to-generalist strategy at the agentic, interactive level rather than at a purely data level, which is common in many previous LLM works.
> Our work validates this approach within the highly diverse and complex domain of GUI interaction, which we see as a critical branch for advancing interactive intelligence. Our results confirm that training specialist agents on different software and then merging them into a single generalist model significantly enhances the agent's universal planning and reflection capabilities, thereby improving overall performance. We will add this clarification to our revised manuscript.
> ***
> ### W2:
> We agree with your assessment. We will revise the manuscript to add more detailed explanations for Section 3.1 and insert appropriate citations for Figure 2 within Section 3. We also acknowledge that Figure 2 is quite dense as it aims to show the complete pipeline; we will simplify and streamline it in the revision to improve readability.
> ***
> ### W3:
> Thank you for pointing this out. This is indeed an error. We will correct the citation to refer to Section A.1 in the revised version.
> ***
> ### Q1:
> Thank you for this suggestion. We do have a concrete example in the supplementary material. Appendix C.1 provides a detailed walkthrough of this process, with the corresponding model inputs and outputs available in Appendix B.1. This example illustrates the full loop of how a curriculum task is automatically proposed and then judged by the WSM within the PowerPoint application.
> ***
> ### Q2:
> A "World Model" in the general sense typically implies a model with the capacity to predict future states. Our "World State Model" (WSM) borrows from this concept, but it is not predictive. Its function is to evaluate the current state and determine if the task has been successfully completed.
> Based on our review of the literature, this specific application and term (WSM as an evaluator) is our own proposal. We would be happy to be corrected if we missed any relevant prior work.
> ***
> We sincerely thank you for your valuable and constructive feedback. We will integrate these insightful comments into our next revision. Unfortunately, the current rating puts us in a difficult position. We are approaching a deadline for another suitable conference, which is forcing us to make an imminent decision about this submission. We sincerely hope our responses are sufficient to address your concerns. We would be very grateful for your timely re-evaluation, as we would strongly prefer to continue with this submission.

---

### Official Review · Reviewer_tCNQ · 2025-10-31

**Soundness:** 2
**Presentation:** 2
**Contribution:** 2
**Rating:** 4
**Confidence:** 5

**Summary:**

This paper proposes SEAgent, a self-evolving framework that enables computer-use agents to autonomously master novel software through curriculum task generation,  World State Model, and experiential learning. But why can a Large Vision-Language Model as a World State Model?

**Strengths:**

The method effectively reduces reliance on human-labeled data by allowing agents to learn from trial-and-error interactions in unfamiliar software environments.

**Weaknesses:**

1. If Large Vision-Language Models (LVLMs) can serve as World State Models, it means that LVLMs contain all the information for the application. So why not directly use LVLMs for decision-making? This completely contradicts the "without human intervention" description in Figure 1's caption and the motivation of this paper.
2. The World State Model relies on GPT-4o annotations, which may introduce bias and affect reproducibility.

**Questions:**

1. If Large Vision-Language Models (LVLMs) can serve as World State Models, it means that LVLMs contain all the information for the application. So why not directly use LVLMs for decision-making? This completely contradicts the "without human intervention" description in Figure 1's caption and the motivation of this paper.
2. The World State Model relies on GPT-4o annotations, which may introduce bias and affect reproducibility.
3. How does the World State Model ensure interpretability in its step-wise trajectory judgments, especially for ambiguous GUI states?
4. Why were certain software applications excluded from ScienceBoard, and how might this affect the validity of OOD performance claims?
5. How does the Curriculum Generator avoid generating repetitive or infeasible tasks as task complexity increases?
6. Can the reward model generalize to software with highly dynamic or non-standard GUI elements not seen during training?

---

> ### Author Response · Authors · 2025-11-13
>
> We thank you for your comprehensive review and insightful feedback on our paper, "SEAgent: A Self-Evolving Framework for Mastering Novel Software." Your comments have highlighted several key areas for clarification and improvement.
>
> Below, we address each of your points in detail.
> ***
> ### Q1-W1:
> We differentiate between the (1) ability to execute a task and the (2) ability to verify its outcome. As you pointed out, these are two separate capabilities. To use an analogy: a person might not know how to perform a complex operation in Excel because they are unfamiliar with it, but this doesn't prevent them from understanding the process of an expert using it and judging whether the operations are correct.
> Our framework leverages this exact distinction. The agent (actor) is like the novice user who needs to learn the procedural, step-by-step "how-to." The World State Model (WSM), powered by an LVLM, acts as the "judge" who can verify the process, even without possessing the procedural knowledge to execute it perfectly itself. Our solution is designed to enable the agent to learn how to use the software through the WSM's feedback. This automated feedback loop replaces the need for human intervention in the learning process.
> ***
> ### Q2-W2:
> We acknowledge this valid concern. To ensure full transparency and reproducibility, we commit to making our entire dataset publicly available upon publication. This includes all GPT-4o annotations used for training the World State Model, as well as our complete training and evaluation code.
> ***
> ### Q3:
> The WSM ensures interpretability by outputting a descriptive natural language caption for each step of the trajectory. This caption explicitly describes the WSM's understanding of the current GUI state and its inference of the actor's action, providing a transparent window into its reasoning.
> ***
> ### Q4:
> The two applications excluded were Lean (a formal prof language) and LaTeX. We made this decision because the primary interaction modality for these applications is heavily text- and code-based, rather than GUI-centric. Our work, SEAgent, is specifically focused on mastering novel software through GUI interaction, and we believe our OOD performance claims remain valid within this defined scope.
> ***
> ### Q5:
> We employ two mechanisms to address this:
> Repetitive Tasks: The Curriculum Generator maintains a context of recently generated tasks to promote diversity (see Appendix D).
> Infeasible Tasks: Our framework is robust to infeasible tasks. If one is proposed, the agent will fail to make progress, and the WSM will assign a low reward. This feedback loop naturally "filters out" such tasks.
> ***
> ### Q6:
> This is an excellent point. Our current model operates on screen representations rather than real-time video streams, largely due to computational constraints. Generalizing to highly dynamic or non-standard GUI elements is a significant challenge. We view this as a very important direction for future research and will explicitly discuss this limitation.
> ***
> We sincerely thank you for your valuable and constructive feedback. We will integrate these insightful comments into our next revision. Unfortunately, the current rating puts us in a difficult position. We are approaching a deadline for another suitable conference, which is forcing us to make an imminent decision about this submission. We sincerely hope our responses are sufficient to address your concerns. We would be very grateful for your timely re-evaluation, as we would strongly prefer to continue with this submission.

---

### Note · Authors · 2025-11-14

**Comment:**

We have decided to withdraw our submission. We sincerely thank the reviewers for their time and valuable opinions. We found their feedback to be very constructive and plan to incorporate their suggestions to make the paper more competitive for a future submission.

**Withdrawal Confirmation:**

I have read and agree with the venue's withdrawal policy on behalf of myself and my co-authors.